# Insights into Synergy of Copper and Acid Sites for Selective Catalytic Reduction of NO with Ammonia over Zeolite Catalysts

**Wenyi Zhao** [1,2], **Menglin Shen** [1,2], **Yueran Zhu** [2], **Xudong Ren** [2] **and Xingang Li** [1,3,*]

[1] Collaborative Innovation Center of Chemical Science and Engineering (Tianjin), Tianjin Key Laboratory of Applied Catalysis Science & Technology, School of Chemical Engineering & Technology, Tianjin University, Tianjin 300350, China

[2] National Key Laboratory for Research and Comprehensive Utilization of Rare Earth Resources in Baiyun Obo, Baotou Research Institute of Rare Earths, Baotou 014030, China

[3] Institute of Shaoxing, Tianjin University, Zhejiang 312300, China

[*] Correspondence: xingang_li@tju.edu.cn; Tel.:+86-22-27403389

**Abstract:** Herein, we report the function of copper sites in Cu-SSZ-13, Cu-ZSM-5 and Cu-Beta catalysts with the same Si/Al ratio (14) and Cu/Al ratio (0.4) on selective catalytic reduction of NO with $NH_3$ ($NH_3$-SCR) and reveal the relationship between active sites (Cu sites, acid sites) and catalytic activity. The results show that the amount of isolated $Cu^{2+}$ ions in the catalysts directly determines the formation of strong Lewis acid sites and reaction intermediate $NO_3^-$ ions, thus affecting the low-temperature SCR performance, while the amount of highly stable $Cu^+$ ions and Brønsted acid sites is related to the high-temperature SCR performance of the catalysts. Consequently, it contains enough isolated $Cu^{2+}$ ions, highly stable $Cu^+$ ions and Brønsted acid sites, which endows Cu-SSZ-13 with excellent $NH_3$-SCR activity.

**Keywords:** $NH_3$-SCR; zeolite; copper cations; acid sites; $NO_3^-$ species





## 1. Introduction

In 2020, the China national automobile nitrogen oxide ($NO_x$) emissions was 6.137 million tons, and the diesel vehicle $NO_x$ emissions had reached 88.8% of the total automobile emissions. Diesel engines are oxygen enriched combustion and the oxygen content in the exhaust is high. It is very difficult to directly carry out a selective catalytic reduction of $NO_x$, which is a worldwide problem. At present, ammonia selective catalytic reduction technology ($NH_3$-SCR) is mainly used to treat the diesel engine $NO_x$ [1]. As $NH_3$-SCR is a dynamic process, reactive gases such as NO, $NO_2$, $O_2$ and $NH_3$ will be adsorbed and activated on the active site to form an active species. The active species and reaction gas are constantly consumed, thus generate important reaction intermediates. Consequently, developing an efficient $NH_3$-SCR catalyst is the key to solving the above issues [2].

With the increasingly strict environmental emission regulations, the $NH_3$-SCR catalyst has been developed from the vanadium tungsten titanium (VWTi) catalyst, which can meet the national IV and V emission standards to copper-based zeolite catalysts that can meet the national VI. In the 1980s, Liu et al. [3,4] first reported that the Cu-ZSM-5 zeolite had good SCR activity for NO decomposition. Consequently, with the gradual deepening of research, more Cu-containing zeolites (such as BEA, MFI, FAU, MOR and CHA etc.) have attracted extensive attention due to their efficient removal of NO [5,6]. However, the Beta zeolite has relatively large 12-rings channels compared with ZSM-5. The research shows that the hydrothermal stability of Cu-Beta zeolite is higher than that of Cu-ZSM-5 with relatively small channel structure [7,8]. The SSZ-13 zeolite has a CHA structure and is a three-dimensional microporous zeolite composed of D6R and CHA composite units alternately.

The copper exchanged SSZ-13 catalyst has become a commercial $NH_3$-SCR catalyst due to it having higher activity and durability than any earlier SCR catalyst [9]. However, with the follow-up implementation of the "China national VI b" and above emission standards, the higher index requirements have been proposed for the low-temperature SCR performance, hydrothermal stability and exhaust by-products ($N_2O$, $NH_3$, etc) of the $NH_3$-SCR catalysts, which has become a current research hotspot. According to the literature [10–12], the nature of active sites such as metal sites and acid sites on the surface of metal-exchanged zeolite catalysts have an important influence on the $NH_3$-SCR reaction. Generally, this can be achieved by tuning the type, number and location of active sites to ameliorate the catalytic activity of the $NH_3$-SCR catalyst. Chen et al. [13] found that the low-temperature SCR activity of Cu-Ce-La-SSZ-13 was enhanced by controlling the position of $Cu^{2+}$ ions, but did not mention that regulating the position of $Cu^{2+}$ ions caused the change of acid sites on the catalyst' surface. Luo et al. [14] indicated through DRIFTS research that $NH_3$ was adsorbed on Lewis acid sites and Brønsted acid sites on the Cu-SSZ-13' surface, but did not point out the relationship between acid sites and catalytic activity in the article. Hence, the roles of copper and acid sites on the surface of copper-based zeolites for the $NH_3$-SCR reaction were compared and analyzed, and the correlation between them and catalytic activity is critical to continue to improve the catalytic performance of the Cu-SSZ-13 catalyst.

In this paper, the intrinsic physicochemical properties of their active sites in the Cu-SSZ-13, Cu-ZSM-5 and Cu-Beta $NH_3$-SCR catalysts with the same Si/Al ratio (14) and Cu/Al ratio (0.4) were comparatively studied, which is crucial to establish the relationship between active site and catalytic performance. We analyzed the reaction processes of $NH_3$ adsorption, NO adsorption and hourly adsorption of NO + $O_2$ and NO + $O_2$ + $NH_3$ in an attempt to understand the effect of active sites on the reaction intermediates formed on the surface of copper-based zeolites in the Standard SCR reaction. The comprehensive characterization results allow us to explain the important reasons for the differences in the $NH_3$-SCR performance among Cu-SSZ-13, Cu-ZSM-5 and Cu-Beta catalysts.

## 2. Results and Discussion

### 2.1. Catalytic Comparison

Figure 1a exhibits the NO conversion as a function of temperature in the standard SCR reaction over Cu-SSZ-13, Cu-ZSM-5 and Cu-Beta catalysts. As shown in Figure 1a, the reaction temperature window (200–500 °C) of the Cu-SSZ-13 catalyst is significantly wider than Cu-ZSM-5 (200–400 °C) and Cu-Beta (300–600 °C), indicating that the Cu-SSZ-13 has the excellent $deNO_x$ performance in the standard SCR reaction. If the reaction temperature is lower than 350 °C, the order of the NO conversion of the Cu based zeolites is Cu-ZSM-5 > Cu-SSZ-13 > Cu-Beta, which suggests that Cu-SSZ-13 and Cu-ZSM-5 catalysts have the superior low-temperature $deNO_x$ activity under the standard SCR conditions. If the reaction temperature is higher than 350 °C, the order of the NO conversion of the Cu based zeolites is Cu-Beta > Cu-SSZ-13 > Cu-ZSM-5. Especially at temperatures higher than 500 °C, the high-temperature SCR performance advantage of the Cu-Beta catalyst is more obvious.

The SCR performance of the Cu-SSZ-13 catalyst before and after hydrothermal aging is shown in Figure 1b. The durability and stability of the catalysts are expressed by the deterioration rate, and this is shown in Figure 1c. In Figure 1c, if the aging temperature is less than 950 °C, the degradation rate of the aged samples does not change significantly with the increase of the aging temperature. After aging at 650 °C, 700 °C and 800 °C, the degradation rates of the samples are 0%, 2% and 1%, respectively. This shows that the Cu-SSZ-13 has a superior hydrothermal stability and durability. If the aging temperature continues to rise to 950 °C, the degradation rate of the aged sample can reach 23.5%. Especially at the reaction temperature higher than 400 °C, the reaction activity of the aged sample is obviously reduced, and the NO conversion at the reaction temperature of 550 °C and above is reduced to zero. This clearly indicates that the Cu-SSZ-13 catalyst still has good catalytic activity and retains the integrity of CHA skeleton structure after high-temperature

hydrothermal aging treatment, but too high aging temperature (≥950°C) will lead to the CHA structure collapse and complete deactivation of the Cu-SSZ-13 catalyst.

**Figure 1.** NO conversion as a function of temperature in the standard SCR reaction over Cu-SSZ-13, Cu-ZSM-5 and Cu-Beta catalysts: (**a**) fresh activity of the Cu based zeolites; (**b**) NO conversion of the Cu-SSZ-13 before and after hydrothermal aging; and (**c**) deterioration rate of the Cu-SSZ-13.

*2.2. Composition and Textural Properties of the Catalysts*

The XRD patterns of Cu-SSZ-13, Cu-ZSM-5 and Cu-Beta catalysts before and after copper ion exchange are presented in Figure 2. The chemical composition of the copper-based zeolites is listed in Table 1. As shown in Table 1, the Cu/Al ratios of the Cu-SSZ-13, Cu-ZSM-5 and Cu-Beta catalysts are 0.40, 0.38 and 0.40, and their corresponding copper ion exchange degrees are 79%, 75% and 80%, respectively. This indicates that the copper ion exchange degree of Cu-SSZ-13, Cu-ZSM-5 and Cu-Beta catalysts prepared by the high-temperature one-time ion exchange method is basically the same, and the Cu loading amounts are 2.4%, 2.3% and 2.5%, respectively. It can be seen from Figure 2a,b that the powder XRD patterns of these catalysts with different zeolite support structures are in excellent agreement with those of zeolite supports (i.e., H-SSZ-13, H-ZSM-5, H-Beta) and no diffraction peak of crystalline phase CuO is detected in all samples ($2\theta = 35.6°$ and $38.7°$), except for minor differences in the relative X-ray peak intensity. The results show that they maintain structural integrity during the copper ion exchange. The Cu species can stably exist in the pore structure system of zeolite supports, and the crystal size of Cu-SSZ-13, Cu-ZSM-5 and Cu-Beta catalysts decreased gradually. The XRD patterns of the copper-based zeolites after the $NH_3$-SCR reaction are presented in Figure 2c. Compared with fresh samples (Figure 2b), after the $NH_3$-SCR reaction, the CHA and BEA structures in Cu-SSZ-13 and Cu-Beta catalysts still exist stably and the peak intensity is obviously enhanced, while the peak intensity of Cu-ZSM-5 is evidently reduced, and its crystal form has a tendency to develop into amorphous state, which may be caused by the collapse of the MFI structure of Cu-ZSM-5, and thus affect its high-temperature SCR performance.

**Table 1.** Chemical composition of the copper-based zeolites.

| Catalyst ID | Cu [1] (wt %) | Si/Al [1] | Cu/Al [1] | Copper Ion Exchange Degree [1] (%) | BET Surface Area (m²·g⁻¹) |
|---|---|---|---|---|---|
| Cu-SSZ-13 | 2.4 | 14 | 0.4 | 79% | 767.3 |
| Cu-ZSM-5 | 2.3 | 14 | 0.4 | 75% | 438.2 |
| Cu-Beta | 2.5 | 14 | 0.4 | 80% | 702.7 |

[1] Determined by elemental analysis.

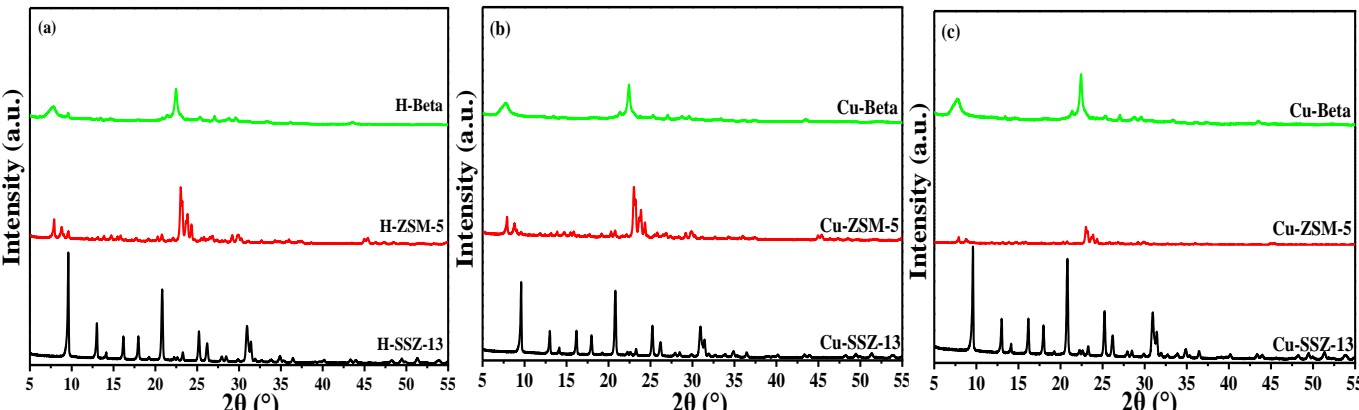

**Figure 2.** XRD patterns of zeolite supports (**a**), copper-based catalysts (**b**) and copper-based zeolites after the NH$_3$-SCR reaction (**c**).

The specific surface area and pore structure parameters of the catalysts with different zeolite support structures are shown in Figure 3a. It can be seen from Figure 4a that the specific surface area and the total pore volume of Cu-SSZ-13, Cu-ZSM-5 and Cu-Beta catalysts are 767.30 m$^2$/g, 0.2835 cm$^3$/g, 438.15 m$^2$/g, 0.1901 cm$^3$/g and 702.68 m$^2$/g, 0.4176 cm$^3$/g, respectively. It is clear that the Cu-SSZ-13 catalyst has the largest specific surface area, while the total pore volume of the Cu-Beta catalyst is relatively larger than Cu-SSZ-13 and Cu-ZSM-5. To some extent, this shows that the larger the specific surface area and the pore volume, the more favorable the contact between the active center and the reactant molecules, which results in an improved catalytic activity [15]. Figure 3b shows the proportion of micropores and mesopores in the Cu based zeolites. As shown in Figure 3b, the order of proportion of microporous volume to total pore volume of the catalysts is Cu-SSZ-13 (94%) > Cu-ZSM-5 (66%) > Cu Beta (49%), indicating that the Cu-SSZ-13 catalyst is a typical microporous material, while Cu-Beta and Cu-ZSM-5 are micro-mesoporous composite materials [16,17], which is consistent with the results shown in Figure S1.

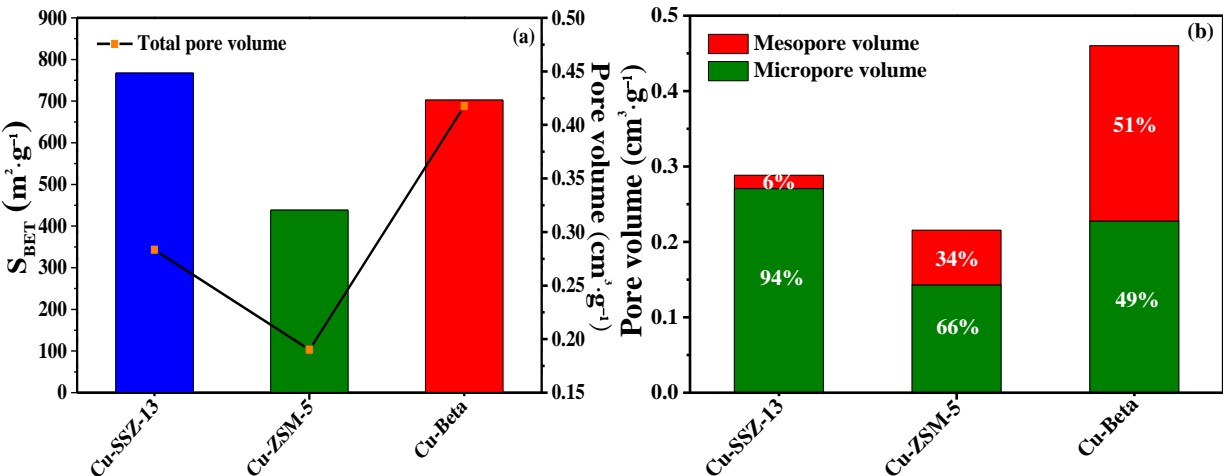

**Figure 3.** Specific surface area and pore structure parameters (**a**) and proportion of micropores and mesopores (**b**) of the copper-based zeolites.

The SEM images of the catalysts are exhibited in Figure 4. As shown in Figure 4a–c, the crystal morphology of the Cu-SSZ-13 catalyst is a regular cubic structure, and the crystal surface is flat and smooth, while Cu-ZSM-5 and Cu-Beta exhibits irregular aggregates and particles adhere to each other. Through a comparison of appearance and particle size test of these catalysts, it is found that the Cu-SSZ-13 sample has a large particle size and excellent mobility, while the Cu-Beta has a relatively small particle size and poor mobility,

and the powder particles are relatively loose and have large electrostatic attraction, which is consistent with the XRD results. In addition, in the $^{27}$Al-NMR signal, the signal peak of non-skeleton aluminum species is not found in Cu-SSZ-13, Cu-ZSM-5 and Cu-Beta catalysts, indicating that all Al atoms in these catalysts exist in the form of skeleton aluminum and the silicon (aluminum) oxygen skeleton structure is relatively stable (Figure S2).

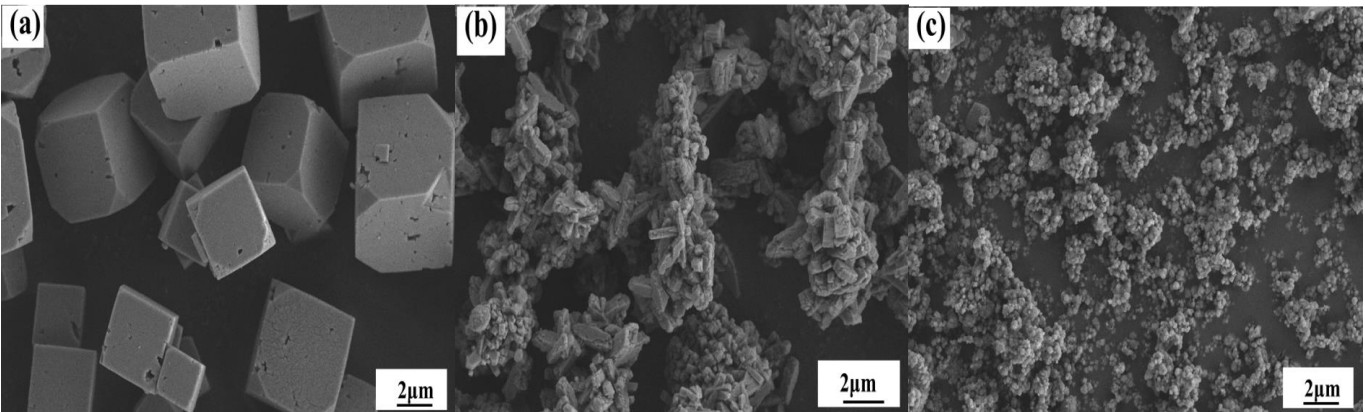

**Figure 4.** SEM images of Cu-SSZ-13 (**a**), Cu-ZSM-5 (**b**) and Cu-Beta (**c**).

### 2.3. Nature of Active Sites

EPR is an effective method of characterization for identifying copper species in the copper-based zeolites. $Cu^{2+}$ ions with one unpaired electron have a strong EPR response, while it should be noted that the Cu species such as $Cu^+$, $[Cu(OH)^+]$ and $CuO_X$ clusters are all EPR-silent [13]. Consequently, the EPR signal of the Cu based zeolites can be attributed to the isolated $Cu^{2+}$. The isolated $Cu^{2+}$ is usually considered as the active site of a $NH_3$-SCR reaction, and its content often determines the activity of the catalysts [18]. To compare the difference in the amount of isolated $Cu^{2+}$ ions in Cu-SSZ-13, Cu-ZSM-5 and Cu-Beta catalysts, all samples were characterized by EPR. Figure 5a presents the EPR spectra for the Cu based zeolites recorded at 110 K. As shown in Figure 5a, Cu-SSZ-13, Cu-ZSM-5and Cu-Beta catalysts all have signal peaks of isolated $Cu^{2+}$ ions at $g_\perp$ = 2.06 and $g_{||}$ = 2.39 (or $g_{||}$ = 2.38), while the negative peaks at high field ($g_\perp$ = 2.06) reflect the axial symmetry of the divalent copper ions [19]. The relative intensity of isolated $Cu^{2+}$ions is obtained by the double integration of the EPR spectra of all samples, and the results are shown in Table 2. The order of the amount of $Cu^{2+}$ ions in the Cu based zeolites is Cu-ZSM-5 ≈ Cu-SSZ-13 > Cu-Beta. It can be seen that the amount of isolated $Cu^{2+}$ in Cu-SSZ-13 and Cu-ZSM-5 catalysts is basically the same, but they are obviously higher than Cu-Beta. The lack of enough isolated $Cu^{2+}$ may also be an important reason why the Cu-Beta catalyst exhibits poor $NH_3$-SCR performance in the low temperature range (150~250 °C). Figure 5b displays the hyperfine structures of copper ions for all samples at low field. The EPR characteristic signal peaks for the Cu based zeolites at $g_{||}$ = 2.39 (or $g_{||}$ = 2.38) should be attributed to $Cu^{2+}$ located in the cationic sites around 6-rings, which is consistent with the reports in the literature [20].

**Table 2.** EPR signal relative intensity for the Cu based zeolites.

| Sample | Relative Intensity (%) |
|---|---|
| Cu-SSZ-13 | 98 |
| Cu-ZSM-5 | 100 [1] |
| Cu-Beta | 43 |

[1] The intensity of the Cu-ZSM-5 sample was defined to be 100%.

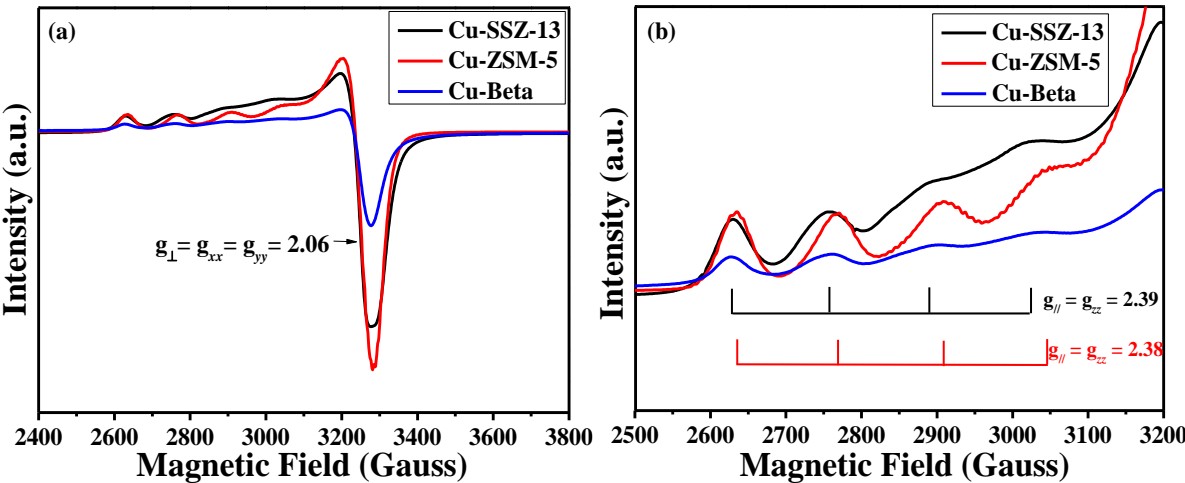

**Figure 5.** EPR spectra recorded at 110 K (**a**) and magnification of the low-field hyperfine structure (**b**) for the copper-based zeolites.

To further understand the copper species status on the surface of the copper-based zeolites, the catalysts' surface were analyzed by X-ray photoelectron spectroscopy (XPS). The XPS spectra reveal that different Cu species existed on the surface of the copper-based zeolites associated with Cu contents. The XPS spectra for Cu 2p are presented in Figure 6. In Figure 6, the XPS spectra of Cu 2p in Cu-SSZ-13, Cu-ZSM-5 and Cu-Beta catalysts have two obvious peaks concentrated at ~952.8 eV and ~933.2 eV, which belong to Cu $2p_{1/2}$ and Cu $2p_{3/2}$, respectively. Through Gaussian fitting, the XPS spectra of Cu 2p of all samples can be fitted with four peaks. The peaks at 953.8~954.4 and 933.9~934.1 eV are attributed to $Cu^{2+}$, the peaks at 952.6~952.9 and 932.9~933.2 eV are assigned to $Cu^+$ [21,22]. Table 3 reveals the relative content of $Cu^+$ and $Cu^{2+}$ ions on the catalysts' surface. The ratio of $Cu^{2+}/(Cu^+ + Cu^{2+})$ was calculated according to the deconvoluted peak areas and used to measure the concentration of the $Cu^{2+}$ species on the surface of the catalyst [23]. According to the area ratio of the Cu peaks, the surface $Cu^{2+}$ concentrations of Cu-SSZ-13, Cu-ZSM-5 and Cu-Beta catalysts were 30%, 29% and 23%, respectively. The indicates that the concentration of $Cu^{2+}$ ions on the surface of Cu-SSZ-13 and Cu-ZSM-5 is significantly higher than Cu-Beta, which may also be an important reason for the better low-temperature SCR activity of Cu-SSZ-13 and Cu-ZSM-5 catalysts. In addition, the ratio of $Cu^+/(Cu^+ + Cu^{2+})$ on the surface of Cu-SSZ-13, Cu-ZSM-5 and Cu-Beta catalysts can all reach more than 70%, so the copper-based zeolites' surface is mainly $Cu^+$ ions. Luo et al. [24,25] believed that more $Cu^+$ ions may increase the amount of oxygen vacancies on the catalysts' surface, which is conducive to the adsorption, activation and migration of oxygen in the gas phase.

**Table 3.** The integral areas of Cu 2p determined from XPS spectra over Cu-SSZ-13, Cu-ZSM-5 and Cu-Beta catalysts.

| Sample | Peak Area (a.u.) | | $Cu^{2+}/Cu^+ + Cu^{2+}$ (%) | $Cu^+/Cu^+ + Cu^{2+}$ (%) |
|---|---|---|---|---|
| | $Cu^{2+}$ | $Cu^+$ | | |
| Cu-SSZ-13 | 10,355 | 24,187 | 30 | 70 |
| Cu-ZSM-5 | 8010 | 19,652 | 29 | 71 |
| Cu-Beta | 6710 | 22,718 | 23 | 77 |

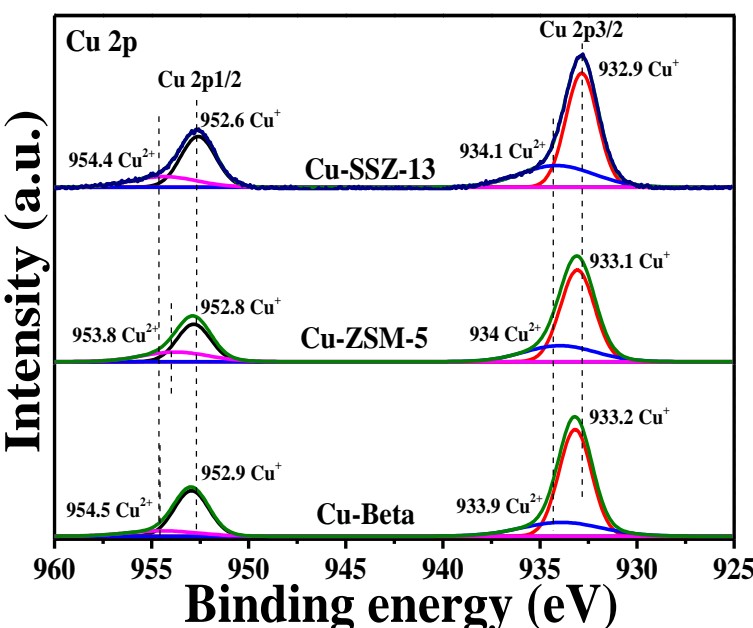

**Figure 6.** XPS spectra of Cu 2p for the copper-based zeolites.

It is generally accepted that the redox properties are an important factor affecting the NH$_3$-SCR reaction. The H$_2$-TPR patterns of the copper-based zeolites are shown in Figure 7a. As shown in Figure 7a, the H$_2$ consumption profiles of Cu-SSZ-13 and Cu-Beta catalysts can be divided into four reduction peaks by Gaussian fitting, while only three reduction peaks can be fitted for the Cu-ZSM-5 catalyst. According to the literature [26,27], the reduction of isolated Cu$^{2+}$ ions located at the ion exchange sites often need to go through two steps. First, the isolated Cu$^{2+}$ ions are reduced to Cu$^+$ ions at a lower temperature, and then the Cu$^+$ ions are completely reduced to Cu$^0$ at a higher temperature. These Cu$^+$ ions include the intermediate products of the two-step reduction of isolated Cu$^{2+}$ ions and the Cu$^+$ ions existing in the catalyst. Therefore, the low-temperature reduction peak in the temperature region 255~333°C can be due to the reduction of the isolated Cu$^{2+}$ to Cu$^+$ [28,29]. The reduction peak of 356~489 °C is attributed to the reduction of the unstable Cu$^+$ to Cu$^0$. The higher-temperature reduction peak at 549~665 °C indicates the reduction of the highly stable Cu$^+$ to Cu$^0$. It can be seen from the above analysis that the reduction temperature of the isolated Cu$^{2+}$ ions in Cu-SSZ-13 catalyst (255 °C) is significantly lower than that of Cu-ZSM-5 (275 °C) and Cu-Beta (333 °C) catalysts, indicating that the reduction of Cu$^{2+}$ to Cu$^+$ in Cu-SSZ-13 is easier. According to Figure 7b, the total H$_2$ consumption of the Cu based zeolites is Cu-SSZ-13 > Cu-ZSM-5 > Cu-Beta, which shows that the Cu-SSZ-13 catalyst has the strongest redox properties among them, especially its low-temperature reduction capacity.

Figure 7c illustrates the proportion of different copper species in the copper-based zeolites. The ratio of various Cu ions to total Cu was calculated according to the H$_2$ reduction peak areas and used to measure the concentration of Cu ions in the catalysts. It can be seen from the Figure 7c that the proportion of the stable Cu$^+$ to Cu in the catalysts is Cu-SSZ-13 (65%) > Cu-Beta (42%) > Cu-ZSM-5 (8%), and the reduction temperature of the stable Cu$^+$ in Cu-SSZ-13 (573~665 °C) and Cu-Beta (614 °C) catalysts is significantly higher than Cu-ZSM-5 (549 °C). This indicates that the high-temperature stability of Cu species in Cu-SSZ-13 and Cu-Beta zeolite frameworks is significantly better than Cu-ZSM-5. It is worth noting that the reduction temperature of the highly stable Cu$^+$ ions is generally considered as the temperature at which the framework structure of zeolite begins to collapse [30,31]. Consequently, this may also be the cause of the poor high-temperature SCR performance of the Cu-ZSM-5 catalyst.

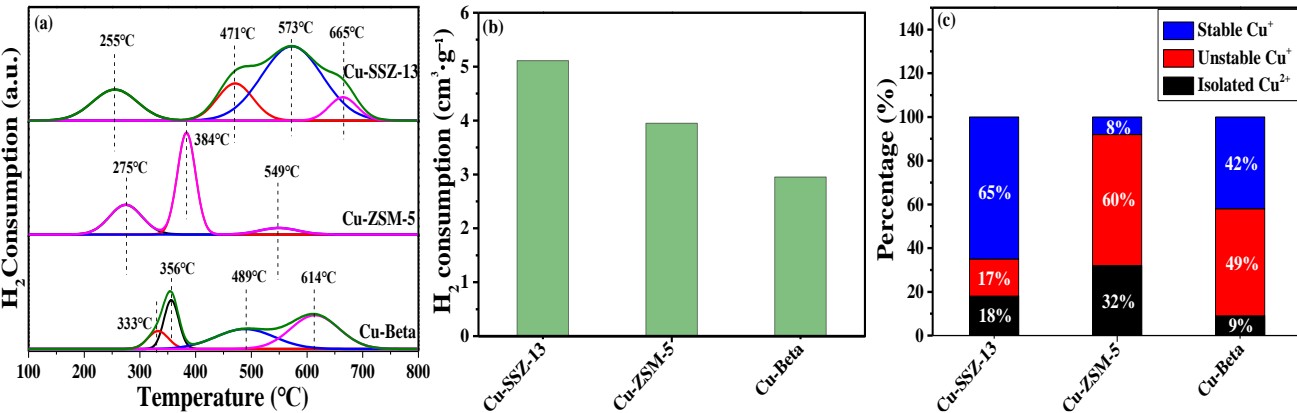

**Figure 7.** H$_2$-TPR patterns (**a**), H$_2$ consumption (**b**) and proportion of different Cu species (**c**) for the copper-based zeolites.

To further understand the local environment of the Cu$^{2+}$ species in the copper-based zeolites, the DRIFTS-NO adsorption experiments were used to explore the nature of the Cu species in the catalysts [32]. The Cu-SSZ-13, Cu-ZSM-5 and Cu-Beta catalysts were exposed to 500 ppm NO in N$_2$ at room temperature for 60 min followed by purging with N$_2$ for 30 min. The NO adsorption properties were obtained by recording the DRIFTS-NO spectra of the catalysts. The DRIFTS spectra of the adsorbed NO on the Cu based zeolites are shown in Figure 8. The absorption peaks at 1950, 1930, 1895 and 1826 cm$^{-1}$ for the catalysts are seen in the spectra. The absorption peak characteristics of Cu-SSZ-13 (1950, 1930 and 1895 cm$^{-1}$), Cu-ZSM-5 (1950 and 1826 cm$^{-1}$) and Cu-Beta (1930 cm$^{-1}$) catalysts are slightly distinct, expressing that there are differences in the location and distribution of copper ions among the three catalytic systems. For the Cu-SSZ-13 catalyst, the peaks at 1950, 1930 and 1895 cm$^{-1}$ correspond to the NO species adsorbed on the two different types of Cu$^{2+}$ ion active sites [33]. The peak at 1950 cm$^{-1}$ is due to the adsorption of NO on the isolated Cu$^{2+}$ ions in the D6R, while the peaks at 1930 and 1895 cm$^{-1}$ are attributed to the NO species adsorbed on the isolated Cu$^{2+}$ ions in the 8-rings window of the CHA cage [34]. Compared with Cu-SSZ-13 catalyst, the Cu-ZSM-5 catalyst appears to show a new absorption peak at 1826 cm$^{-1}$ in addition to a peak at 1930 cm$^{-1}$, which is attributed to the NO species adsorbed on Cu$^+$ ions [35], while the peak of Cu-ZSM-5 and Cu-Beta at 1930 cm$^{-1}$ represents the interaction between NO and isolated Cu$^{2+}$ [36]. Therefore, it can be seen from the above analysis that NO is mainly adsorbed on isolated Cu$^{2+}$, thus forming the Cu$^{2+}$-NO active species. In addition, the order of the NO absorption peak intensity in the 1800–2000 cm$^{-1}$ region is Cu-ZSM-5 ≈ Cu-SSZ-13 > Cu-Beta, proclaiming that the adsorption capacity of Cu-ZSM-5 and Cu-SSZ-13 catalysts for NO is significantly better than Cu-Beta. According to the EPR analysis results, the amount of isolated Cu$^{2+}$ions in Cu-ZSM-5 and Cu-SSZ-13 catalysts is more than Cu-Beta, which is also an important reason for their strong NO adsorption capacity.

In situ DRIFTS-NH$_3$ adsorption experiments were carried out to examine the NH$_3$ adsorption properties of the catalysts [37]. The Cu-SSZ-13, Cu-ZSM-5 and Cu-Beta catalysts were exposed to 500 ppm NH$_3$ in N$_2$ at room temperature for 60 min followed by purging with N$_2$ for 30 min. The NH$_3$ adsorption properties were obtained by recording the DRIFTS-NH$_3$ spectra of the catalysts. Figure 9 displays the difference DRIFT spectra in the zeolite framework region of the copper-based catalysts with different zeolite support structures. In the N-H bonds stretching region, there are three strong absorption peaks related to NH$_3$ adsorption at 3000–3400 cm$^{-1}$, which are the characteristic peak of NH$_4^+$ ions (3350 and 3275 cm$^{-1}$) and coordinated ammonia (3180 cm$^{-1}$) [38]. In the N-H bonds bending vibration region, 1625 and 1255 cm$^{-1}$ can be assigned to the NH$_3$ coordinated to the Lewis acid sites, while 1480 cm$^{-1}$ is due to the bending vibration of NH$_4^+$ ions on Brønsted acid sites [39]. Hence, the ammonia adsorbed on Brønsted acid sites were

much stronger than that on Lewis acid sites. In addition, there are two negative bands in the 3500–3800 cm$^{-1}$ and 800–1000 cm$^{-1}$ regions. According to the literature report [40], 3780, 3675, 3625 and 3540 cm$^{-1}$ are the characteristic peaks of the O-H bonds. Among them, 3780, 3675 and 3625 cm$^{-1}$ are caused by $NH_3$ adsorbed on Si-OH, Cu-OH and Al-OH, respectively [41–43], and 3540 cm$^{-1}$ corresponds to the bending vibration of the Si-OH-Al bond, which is a feature of Brønsted acid sites on the zeolite skeleton [44]. In the 800–1000 cm$^{-1}$ region, it represents the characteristic peaks on the copper ion sites. The Cu-O bond formation in copper-exchanged zeolites can perturb the T-O-T vibrations of the zeolite support framework, thus being useful in the identification of different types of intrazeolitic Cu sites [45,46]. KWAK et al. [47] believe that $NH_3$ can be adsorbed on two types of copper ions in Cu-SSZ-13, corresponding to two characteristic peaks of 940 and 875 cm$^{-1}$, respectively. The first 940 cm$^{-1}$ is attributed to the $NH_3$ adsorbed on the isolated copper ions in the CHA cage, while 875 cm$^{-1}$ is due to the adsorption of $NH_3$ on the copper ions in the 6-MR. It is well known that the Brønsted acid sites are substances that can give protons ($H^+$), while the Lewis acid sites can accept electron pairs. Consequently, compared to Cu-SSZ-13, similar features of ammonia adsorption are also observed during the adsorption process on Cu-ZSM-5 and Cu-Beta. Four Bønsted acid sites exist on the zeolite catalysts' surface, namely Si-OH, Cu-OH, Al-OH and Al-OH-Si, and the Lewis acid sites are mainly two types of Cu exchange sites, which are isolated copper ions in CHA cage and 6-MR.

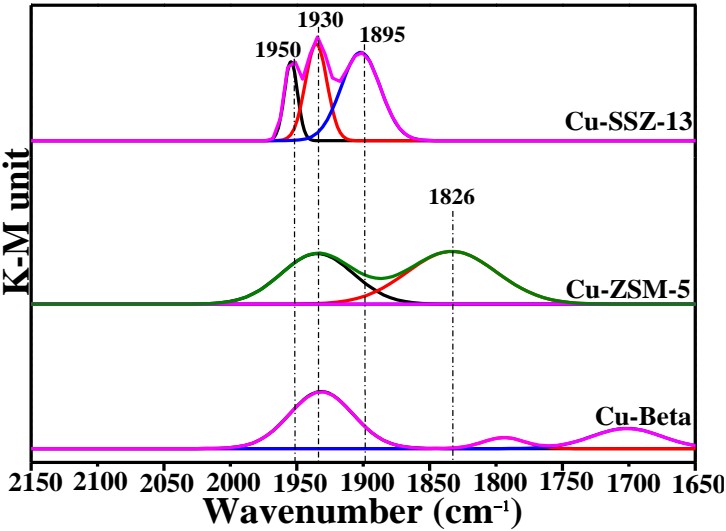

**Figure 8.** DRIFT Spectra of NO adsorption on the copper-based zeolites (Prior to NO adsorption, each catalyst was oxidized at 500 °C for 60 min).

Surface acidity is one of the most important parameters that evaluates the extent of NO reduction with $NH_3$ over copper-based zeolites, and it is usually used to determine the acid sites of the zeolite material for the adsorption and activation of $NH_3$, and the type and quantity of the acid sites will affect the catalytic activity [48,49]. Figure 10a shows the $NH_3$-TPD profiles of the Cu based zeolites. Four $NH_3$ desorption peaks appear for the Cu based zeolites: the low-temperature A and B peaks at 126~196 °C, the medium-temperature C peak at about 370 °C and the high-temperature D peak above 500 °C, which correspond to the four types of acid sites in the zeolite catalysts [50,51]. Among them, the desorption peaks (A and B peaks) before 200 °C are attributed to the weak adsorption of $NH_3$, mainly including physical adsorbed $NH_3$ (A peak) and $NH_3$ chemically adsorbed on the weak Lewis acid sites (B peak). The C peak at about 370 °C is assigned to the $NH_3$ adsorbed on the strong Lewis acid sites, while the D peak above 500 °C is related to $NH_3$ adsorbed on the Brønsted acid sites. According to the literature report [52] and $NH_3$-DRIFT analysis results, four Brønsted acid sites exist on the zeolite catalysts' surface, namely Si-OH, Cu-OH,

Al-OH and Al-OH-Si, and the Lewis acid sites are mainly two types of Cu ion exchange sites, which are isolated copper ions in CHA cage and 6-MR.

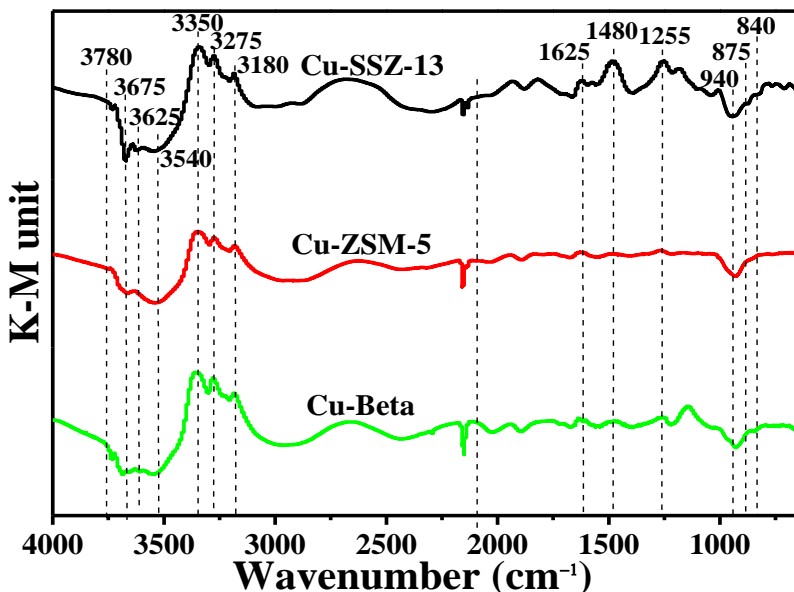

**Figure 9.** DRIFTS Spectra of $NH_3$ adsorption on the copper-based zeolites (Prior to $NH_3$ adsorption, each catalyst was oxidized at 500 °C for 60 min).

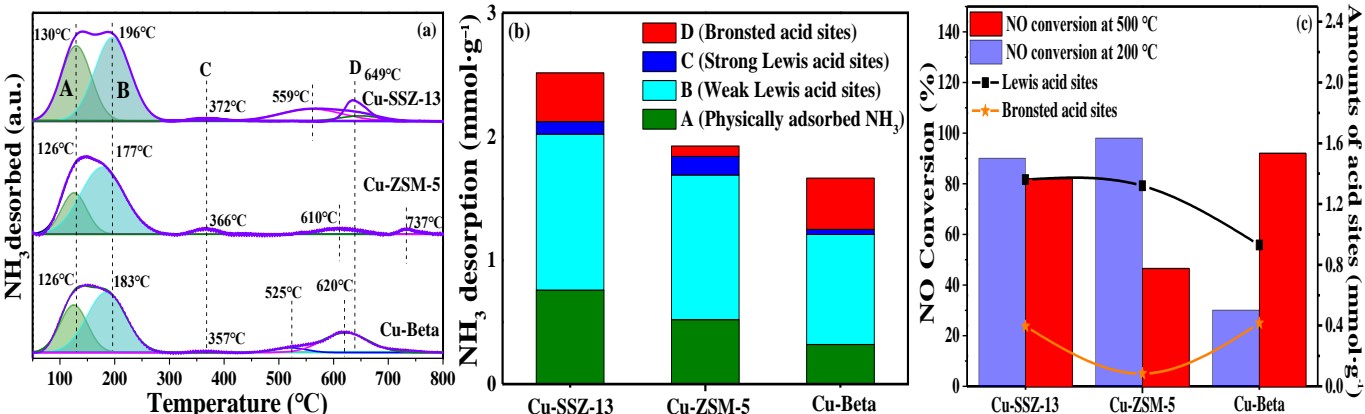

**Figure 10.** $NH_3$-TPD patterns (**a**), amount of $NH_3$ desorption (**b**) and relationship between acid sites and catalytic activity (**c**) for the copper-based zeolites.

After quantitative calculation of the peak separation fitting, the acid site distribution and the acid amount (desorption amount of $NH_3$) in the Cu based zeolites are shown in Figure 10b. The order of total acidity of the Cu based zeolites is Cu-SSZ-13 (2.61 mmol·g$^{-1}$) > Cu-ZSM-5 (2.13 mmol·g$^{-1}$) > Cu-Beta (2.05 mmol·g$^{-1}$), indicating that more acid sites can be generated on the Cu-SSZ-13' surface to adsorb $NH_3$. Moreover, the amount of Lewis acid sites in Cu-SSZ-13 (1.36 mmol·g$^{-1}$) and Cu-ZSM-5 (1.32 mmol·g$^{-1}$) is clearly higher than in Cu-Beta (0.93 mmol·g$^{-1}$), illustrating that the amount of isolated $Cu^{2+}$ ions in Cu-Beta is the least, which has been confirmed in the EPR results. For the Brønsted acid site, if the reaction temperature exceeds 500 °C, the $NH_3$ adsorbed on the acid site can be released. Thus, the more Brønsted acid sites in the catalysts, the stronger its storage capacity for $NH_3$. Compared with Cu-SSZ-13 (0.40 mmol·g$^{-1}$) and Cu-Beta (0.42 mmol·g$^{-1}$), the amount of Brønsted acid sites in the Cu-ZSM-5 (0.08 mmol·g$^{-1}$) is the lowest, explaining that the Cu-ZSM-5 has the weakest $NH_3$ storage capacity at high temperature. Figure 10c reveals the relationship between acid sites and catalytic activity. It can be seen from the figure that the amount of Lewis acid sites in the zeolite catalysts is in the same order as the

low-temperature SCR activity (200 °C), while the amount of Brønsted acid sites is related to the high-temperature SCR activity (500 °C). Consequently, the strong Lewis acid sites may be beneficial to the low-temperature SCR reaction, while the Brønsted acid sites perhaps promote the high-temperature SCR reaction. The Cu-SZZ-13 contains enough Lewis and Brønsted acid sites, so it shows the best catalytic activity.

### 2.4. Reaction Intermediates

To more completely understand the differences in the catalytic properties of Cu-SSZ-13, Cu-ZSM-5 and Cu-Beta catalysts for $NH_3$-SCR, the formation of reaction intermediates over these catalysts during exposure to a mixture of 500 ppm NO and 10% $O_2$ ($N_2$ as balance gas) at 150 °C was studied by in situ DRIFT spectra. As shown in Figure 11a–c, when the Cu-SSZ-13 catalyst is exposed to the mixed gas of NO and $O_2$ for up to 30 min, there is a continuous increase in the intensity of the absorption peak in the 1250–1750 $cm^{-1}$ region, which is resolved into three peaks at 1625, 1575 and 1500 $cm^{-1}$. These peaks can be attributed to nitrate ions ($NO_3^-$) gradually generated on the Cu sites with increasing exposure time to 30 min, while the peak round 1575 $cm^{-1}$ can be assigned to bidentate nitrate [53], and the amount of nitrate ion will increase with the increase of exposure time. A similar trend can be observed in the in situ DRIFT spectra of Cu-ZSM-5 and Cu-Beta catalysts compared to the Cu-SSZ-13, except for the latter two only appear two peaks around 1625 and 1575 $cm^{-1}$. According to DRIFTS-NO adsorption results, NO is mainly adsorbed on isolated $Cu^{2+}$ ions and forms the $Cu^{2+}$-NO active species. Hence, it can be considered that $NO_3^-$ is also mainly formed on the active $Cu^{2+}$ ion sites. The intensity sequence of $NO_3^-$ absorption peaks in the 1250–1750 $cm^{-1}$ region is Cu-ZSM-5 > Cu-SSZ-13 > Cu-Beta, indicating that more $NO_3^-$ will be formed on the Cu-ZSM-5' surface, which may be caused by more isolated $Cu^{2+}$ contained in the Cu-ZSM-5 catalyst. Therefore, $NO_3^-$ is known as one of the key reaction intermediates for the $NH_3$-SCR reaction over metal-exchanged zeolite catalysts.

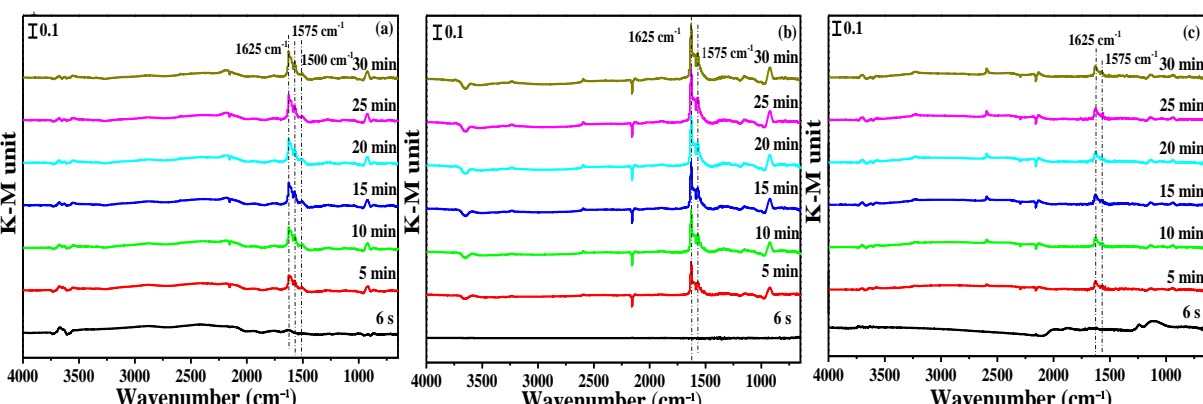

**Figure 11.** In situ DRIFT spectra of NO + $O_2$ adsorption on the copper-based zeolites at 150 °C: (**a**) Cu-SSZ-13; (**b**) Cu-ZSM-5 and (**c**) Cu-Beta.

The mixture of NO + $O_2$ is pre-adsorbed on the copper-based zeolites' surface at 150 °C for 30 min to generate $NO_3^-$ followed by turning off NO + $O_2$ and purging with $N_2$ for 30 min. Figure 12 shows the in situ DRIFT spectra of $NH_3$ reacted with pre-adsorbed NO + $O_2$ on the copper-based zeolites at 150 °C for 30 min. The DRIFT spectra with a reaction time of 0 min is recorded after exposure to a mixture of 500 ppm NO and 10% $O_2$ ($N_2$ as balance gas) at 150 °C for 30 min followed by purging with $N_2$ for 30 min. It is easy to see in Figures 11 and 12 that no noticeable changes in the DRIFT spectra of the catalysts were caused by $N_2$ purging for 30 min after $NO_3^-$ formation. The intensity of the $NO_3^-$ absorption peak of the catalysts in the 1250–1750 $cm^{-1}$ region will gradually weaken with the increase of exposure time in $NH_3$ until it disappears. Among them, the $NO_3^-$ absorption peaks of Cu-SSZ-13 and Cu-ZSM-5 catalysts at 1625 and 1500 $cm^{-1}$ disappear after 20 min exposure

in NH₃, while the NO₃⁻ absorption peaks of Cu-Beta catalyst disappears completely after 15 min. This indicates that the rate of NO₃⁻ disappearance is much faster in Cu-Beta than in Cu-SSZ-13 and Cu-ZSM-5. Simultaneously, two new peaks appear in Cu-SSZ-13 and Cu-ZSM-5 catalysts around 1620 and 1470 cm⁻¹, while the Cu-Beta catalyst only has a weak absorption peak at 1620 cm⁻¹. These two peaks are attributed to the adsorption of NH₃ onto the strong Lewis acid sites generated by the exchange of isolated $Cu^{2+}$ ions and the Brønsted acid sites generated by zeolite protons, respectively [54]. This distinctly shows that the NO₃⁻ species produced in Cu-SSZ-13, Cu-ZSM-5 and Cu-Beta catalysts can be reduced by NH₃ [55]. It is worth noting that the disappearance rate of the reaction intermediates NO₃⁻ in the Cu based zeolites is Cu-Beta > Cu-SSZ-13 > Cu-ZSM-5, which may be due to the fact that the concentration of the NO₃⁻ ions generated in the Cu-Beta catalyst is less than in Cu-SSZ-13 and Cu-ZSM-5, rather than its higher reactivity with NH₃, which has been well confirmed in Figure 11. Consequently, the most likely reason is that the amount of isolated $Cu^{2+}$ ions in the Cu-Beta catalyst is the least compared to Cu-SSZ-13 and Cu-ZSM-5 catalysts, which leads to the weakest adsorption capacity of Cu-Beta for NO, thus affecting the amount of the $Cu^{2+}$-NO active species, and ultimately limiting the formation of the NO₃⁻ species that play an important role in the low-temperature SCR reaction. This well explains that the standard SCR activity of the Cu-Beta catalyst at low temperature (≤300 °C) is significantly lower than Cu-SSZ-13 and Cu-ZSM-5.

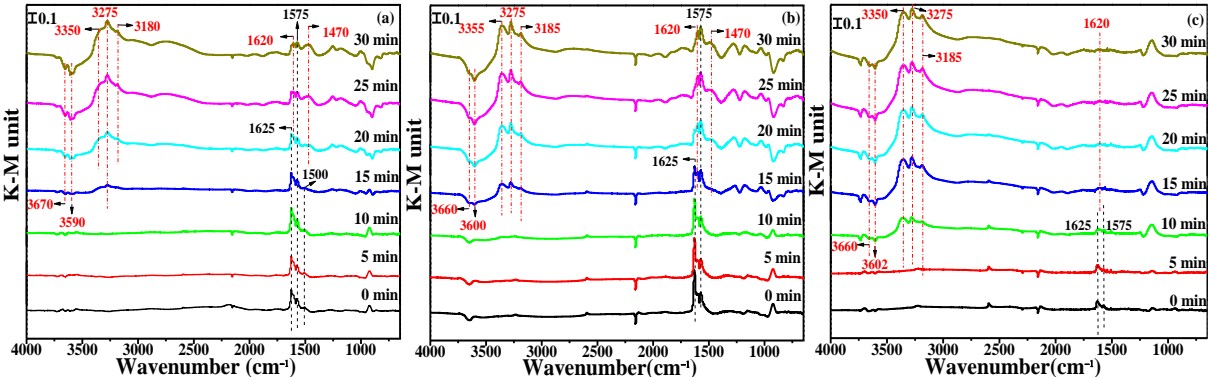

**Figure 12.** In situ DRIFT spectra of NH₃ reacted with pre-adsorbed NO + O₂ on the copper-based zeolites at 150 °C for 30 min: (**a**) Cu-SSZ-13; (**b**) Cu-ZSM-5 and (**c**) Cu-Beta.

In addition, it can be observed from Figure 12 that the Cu-SSZ-13, Cu-ZSM-5 and Cu-Beta catalysts appear as two negative absorption peaks in the 3500–3700 cm⁻¹ region and as three positive absorption peaks with high peak intensity in the 3000–3400 cm⁻¹ region after exposure to NH₃ for 10 min. According to DRIFTS-NH₃ analysis results, the two negative absorption peaks at 3500–3700 cm⁻¹ can be attributed to the weak NH₃ adsorption of OH bands on the Cu-based zeolites' surface, while the three positive absorption peaks at 3000–3400 cm⁻¹ are ascribed to the vibration of the N-H bonds. Therefore, it can be seen from the above analysis that the reaction intermediates NO₃⁻ produced in the Cu-SSZ-13, Cu-ZSM-5 and Cu-Beta catalysts are reduced by NH₃ until it disappears, the excess NH₃ mainly exists on the catalysts' surface in the form of the weak NH₃ adsorption at low temperature.

## 2.5. Effect of Copper and Acid Sites on NH₃-SCR Performance

From the above analysis results, it can be seen that different zeolite skeleton structures can affect the amount of isolated $Cu^{2+}$ ions, the distribution of Cu species, the surface acidity and the amount of reaction intermediates in the catalysts, thus determining the catalytic activity and hydrothermal stability of the copper-based zeolites. First of all, Cu-SSZ-13 and Cu-ZSM-5 have similar catalytic activities in the low temperature region (150~300 °C), and their NO conversion is obviously better than Cu-Beta in this region. However, the Cu-ZSM-5 has the lowest catalytic activity in the high temperature region (400~600 °C),

especially as the reaction temperature reaches 600 °C, its NO conversion can be reduced to below 15%. Secondly, the chemical state, distribution and amount of the Cu species are one of the important reasons for the difference of $NH_3$-SCR performance of the zeolite catalysts. From the XPS and $H_2$-TPR results, it can be found that there are two types of copper cations in these catalysts: isolated $Cu^{2+}$ and $Cu^+$, and the latter one mainly locates on the catalyst ($Cu^+/Cu > 70\%$). Although the content of the isolated $Cu^{2+}$ ions in the copper-based zeolites is small, the reduction temperature of $Cu^{2+}$ ions is significantly lower than $Cu^+$ ions, indicating that $Cu^{2+}$ ions are easier to reduce than $Cu^+$ ions. Therefore, the isolated $Cu^{2+}$ ions are generally considered as the active sites of the $NH_3$-SCR reaction, and their content often has a significant impact on the low-temperature SCR activity of the catalyst [18]. Moreover, the Cu content in the Cu-SSZ-13, Cu-ZSM-5 and Cu-Beta catalysts is basically the same (2.3~2.5%), but the EPR and XPS results show that the amount of isolated $Cu^{2+}$ ions on the skeleton or surface of Cu-Beta is the least, suggesting that the microporous structure (i.e., pore size) of the copper-based zeolites has an impact on the amount of isolated $Cu^{2+}$ ions in the catalyst. Consequently, the lack of enough isolated $Cu^{2+}$ ions is one of the reasons for the poor SCR activity of Cu-Beta in the low temperature region (150~300 °C). For $Cu^+$ ions, it can be found from the $H_2$-TPR results that there are highly stable $Cu^+$ ions that can be reduced at high temperatures (>500 °C) and unstable $Cu^+$ ions with relatively low reduction temperatures (<500°C) in Cu-zeolites. Gao et al. [30,31] believed that the reduction temperature of highly stable $Cu^+$ ions is usually the temperature at which the framework structure of zeolite begins to collapse. According to the results in Figure 7c, the proportion of stable $Cu^+$ to Cu in Cu-ZSM-5 (8%) is significantly lower than Cu-SSZ-13 (65%) and Cu Beta (42%). Hence, this may also be one of the reasons for the poor high-temperature SCR performance of Cu-ZSM-5. Thirdly, the type and number of acid sites are also the key factors affecting the catalytic activity. It can be seen from the $NH_3$-TPD results that there are mainly two types of acid sites on the surface of the copper-based zeolites, namely Lewis and Brønsted acid sites. The total acidity of the Cu-SSZ-13 catalyst is significantly higher than Cu-ZSM-5 and Cu-Beta, and the differences in the physical adsorbed ammonia related to the pore structure of these three catalysts is the main reason for the difference in the total acidity. Compared with Cu-SSZ-13 and Cu-ZSM-5, Cu-Beta has the least number of the Lewis acid sites generated by isolated $Cu^{2+}$ ions, which further proves that the amount of isolated $Cu^{2+}$ ions in Cu-Beta is the least, thus affecting its low-temperature SCR performance. For the Brønsted acid site, Wang et al. [56] found that if the reaction temperature exceeds 500 °C, the $NH_3$ adsorbed on the acid site can be released, which plays an important role in storing and providing $NH_3$ reaction molecules to the active Cu sites in the $NH_3$-SCR reaction. According to Figure 10b, the amount of Brønsted acid sites in Cu-ZSM-5 is apparently lower than Cu-SSZ-13 and Cu-Beta. Therefore, the lack of sufficient Brønsted acid sites may also be the cause of poor high-temperature SCR performance of Cu-ZSM-5 [57,58]. Finally, $NO_3^-$ is known as one of the key reaction intermediates for the $NH_3$-SCR reaction over metal-exchanged zeolite catalysts. According to the in situ DRIFT spectra analysis results, the amount of isolated $Cu^{2+}$ ions in the copper-based zeolites is the basic reason to limit the effective formation of $NO_3^-$ ions, which play an important role in the low-temperature SCR reaction. This well explains that the standard SCR activity of the Cu-Beta catalyst at low temperature ($\leq 300$ °C) is clearly lower than Cu-SSZ-13 and Cu-ZSM-5. In conclusion, the amount of isolated $Cu^{2+}$ ions in the catalysts directly determines the formation of Lewis acid sites and reaction intermediate $NO_3^-$ ions, thus affecting the low-temperature SCR performance, while the amount of highly stable $Cu^+$ ions and Brønsted acid sites is related to the high-temperature SCR performance of the catalysts.

## 3. Materials and Methods

### 3.1. Catalyst Preparation

Cupric acetate ($Cu(CH_3COO)_2 \cdot H_2O$) and H-SSZ-13 (Weihai Baidexin New Materials Co., Ltd., Weihai, China), H-ZSM-5 (Tianjin Nanfang Chemical Catalyst Co., Ltd., Tianjin,

China) and H-Beta (Tianjin Baotou Steel Rare Earth Research, Tianjin, China) commercial supports with the Si/Al ratio of 14 were used for the experiment. The purity of cupric acetate produced by Tianjin Fengchuan Chemical Reagent Technology Co., Ltd. (Fengchuan, Tianjin, China) was 99%, and the specific surface areas of H-SSZ-13, H-ZSM-5 and H-Beta zeolites were 805.83, 474.04 and 759.91 $m^2 \cdot g^{-1}$, respectively. A certain amount of the hydrogen-type zeolite was slowly added into a 0.01 mol $L^{-1}$ copper acetate solution according to the ratio of solid to liquid of 1:50 (g:mL) during stirring, and then ion exchanged in a water bath at 90 °C for 6 h, filtered, washed until the solution reached neutral, dried overnight at 90 °C and calcined at 550 °C for 8 h to obtain the copper-based catalysts with different zeolite supports. The fresh catalyst was thermally aged in 10% $H_2O$ in air at 650 °C for 100 h, 700 °C for 50 h, 800 °C for 16 h and 950 °C for 2 h, respectively, and labeled as aged-650 °C-100 h, aged-700 °C-50 h, aged-800 °C-16 h and aged-950 °C-2 h. The durability of the catalyst was expressed using the deterioration rate, which is the average value for the decrease in NO conversion for the aged samples at four temperature test points (200, 250, 450 and 500 °C).

*3.2. Evaluation of the Catalytic Activity*

The activity of 0.06 g of the catalysts was tested in a fixed-bed quartz reactor (inner diameter = 6 mm). The gas mixture simulates a real diesel exhaust, which contains 500 ppm NO, 500 ppm $NH_3$, 8 vol% $CO_2$, 10 vol% $O_2$, 5 vol% $H_2O$ and $N_2$ as the balance gas. The total flow rate was 300 mL·min$^{-1}$ corresponding to a gas hourly space velocity (GHSV) of 300,000 h$^{-1}$. The effluent gas, including NO, $NO_2$ and $O_2$ was continuously analyzed using an online Fourier infrared analyzer. The results for the steady-state activity were collected after 20 min at each temperature. The NO conversion was calculated as follows:

$$\text{NO conversion} = \frac{[NO]_{inlet} - [NO]_{outlet}}{[NO]_{inlet}} \times 100\%$$

*3.3. Characterization of the Catalysts*

X-ray powder diffractometry (XRD) analysis was performed on a panalytical X-ray powder diffraction analyzer (PANalytical B.V., Almelo, The Netherlands). A Cu Ka ($\lambda$ = 1.5406 Å) was used as the radiation source. The test conditions used were a tube current of 40 mA and a tube voltage of 40 kV. During the test, wide-angle XRD scanning was performed at a speed of 2°/min in the range of 2θ = 5–55° with a step size of 0.02°.

The Brunauer—Emmett—Teller analysis (BET) and pore structure distribution were measured at −196 °C on a 3H-2000PM2 physical adsorption instrument manufactured by the Bester company (Beijing, China) by using the nitrogen adsorption–desorption method. The surface area and pore size distribution curve were determined using the density functional theory (DFT) method in the 0–0.3 partial pressure range. Before the test, the sample was desorbed in a vacuum at 200 °C for 3 h.

Microstructures of the catalyst samples were observed with a sigma 500 electron microscope of ZEISS (Carl Zeiss, Oberkochen, Germany). The test conditions used were a voltage of 3 KV and a working distance of 6 mm.

X-ray photoelectron spectroscopy (XPS) analysis was obtained using a Thermo ESCALAB 250Xi spectrometer (ThermoFisher Scientic, Waltham, MA, USA) with Al Ka radiation (1486.6 eV). The binding energy (B.E.) spectrum was calibrated according to the C 1s standard spectrum (B.E. = 284.6 eV). The composition on the surface of the catalyst according to the atomic ratios was calculated, and the Shirley background and Gaussian—Lorentzian methods were used for peak analysis $^{29}$Si and $^{27}$Al MAS NMR spectra (Bruker, Billerica, MA, USA) were recorded using a 4 mm-diameter zirconia rotor on a Bruker AVANCE III 400 MHz WB operating at 79.5 MHz and 104.3 MHz, respectively. The rotor was spun at 12 kHz for $^{29}$Si and $^{27}$Al MAS NMR and using single pulse sampling. $SiO_2$ and $Al(OH)_3$ were used as chemical shift references for $^{27}$Al MAS NMR and $^{29}$Si MAS NMR, respectively.

Electron paramagnetic resonance (EPR) measurements were obtained at 110 K using a Bruker EMX-10/12-type spectrometer in the X-band (Bruker, Billerica, MA, USA). The dehydrated sample was prepared by placing 50 mg of the fresh sample in a quartz tube and pretreating under dry $N_2$ at 550 °C for 3 h. The sample was then cooled and sealed for measurement. The choice of a lower measuring temperature (110 K) can effectively avoid signal broadening and loss caused by coupling between copper ions.

$H_2$ temperature programmed reduction ($H_2$-TPR) was measured using a Quantachrome: Chem BET chemisorption analyzer (Micromeritics, Norcross, GA, USA) and a thermal conductivity detector (TCD) detector was used. Before the test, 80 mg of the sample was heated from room temperature to 300 °C at a rate of 10 °C· $min^{-1}$ for the drying pretreatment and purged with He gas flow (30 mL/min) for 1 h to remove the impurities adsorbed on the sample. Then the sample was cooled to 30 °C, and the gas flow was switched to 10% $H_2$/Ar mixture (30 mL/min). After the baseline was stabilized for 0.5 h, the sample was heated from room temperature to 800 °C at a rate of 10 °C·$min^{-1}$. The signal is recorded through TCD, and the $H_2$ consumption is calculated from the peak area.

$NH_3$ temperature programmed desorption ($NH_3$-TPD) was conducted on a Quantachrome: Chem BET chemisorption analyzer (Micromeritics, Norcross, GA, USA). First, 80 mg of the sample was pretreated at 300 °C in an He flow at 30 mL·$min^{-1}$ for 2 h, then cooled down to 50 °C. It was adsorbed in 5 vol% $NH_3$/He for 30 min (20 mL·$min^{-1}$). After that, the $NH_3$ was turned off, the gas flow was switched back to pure He (20 mL·$min^{-1}$) again and it was purged for 30 min to remove the physically adsorbed species. Then, the desorption was completed by increasing the temperature from 50 to 800 °C at a rate of 10 °C·$min^{-1}$. After calibration with standard $NH_3$, the desorption amount of $NH_3$ through the peak area was calculated.

The in situ diffuse reflectance infrared transform spectroscopy (in situ DRIFTS) was applied on a Nicolet 6700 FTIR spectrometer (ThermoFisher Scientic, Waltham, MA, USA) with an in situ diffuse reflectance pool and a highly sensitive MCT detector cooled by liquid $N_2$. Prior to IR measurements, a sample of 10 mg mounted in a ceramic holder was heated at 500°C under 21% $O_2$ in $N_2$ flow for 60 min and cooled to the desired temperature. Then, the difference DRIFT spectra of adsorbed $NH_3$ or NO were obtained after exposure of 500 ppm $NH_3$ or 500 ppm NO in $N_2$ flow at room temperature for 60 min followed by purging with $N_2$ for 30 min, respectively. For in situ DRIFT experiments, the difference DRIFT spectra were recorded as a function of exposure time during the stepwise introduction of reactant gases (e.g., first 500 ppm NO and 10% $O_2$ and then 500 ppm $NH_3$) balanced with $N_2$ at 150 °C, respectively.

## 4. Conclusions

In this paper, the $NH_3$-SCR performance differences of Cu-SSZ-13, Cu-ZSM-5 and Cu-Beta catalysts prepared by the ion-exchange method were systematically studied. It is found that there are two copper cations (isolated $Cu^{2+}$ and $Cu^+$) and two acid sites (Lewis and Brønsted acid sites) in these catalysts, and they play different catalytic roles in $NH_3$-SCR reaction. The amount of isolated $Cu^{2+}$ ions in the catalysts directly determines the formation of Lewis acid sites and reaction intermediate $NO_3^-$ ions, thus affecting the low-temperature SCR performance; while the amount of highly stable $Cu^+$ ions and Brønsted acid sites is related to the high-temperature SCR performance of the catalysts. The Cu-SSZ-13 catalyst contains enough isolated $Cu^{2+}$ ions, highly stable $Cu^+$ ions and Brønsted acid sites, so it has excellent $NH_3$-SCR performance. However, the Cu-ZSM-5 has the worst catalytic activity in the high temperature region (400–600 °C) due to the lack of sufficient stable $Cu^+$ ions and Brønsted acid sites. For Cu-Beta, the lack of enough isolated $Cu^{2+}$ ions will lead to a significant reduction in the formation of Lewis acid sites and reaction intermediates $NO_3^-$ ions, making it the worst catalytic activity in the low temperature region (150–300 °C).

**Supplementary Materials:** The following supporting information can be downloaded at: https://www.mdpi.com/article/10.3390/catal13020301/s1, Figure S1: $N_2$ adsorption-desorption isotherms of the copper-based zeolites; Figure S2: $^{27}$Al (a) and $^{29}$Si (b) NMR Spectra of the copper-based zeolites.

**Author Contributions:** Conceptualization, W.Z.; formal analysis, W.Z.; investigation, W.Z., Y.Z.; resources, M.S.; data curation, W.Z.; writing—original draft preparation, W.Z.; writing—review and editing, X.L.; supervision, X.R.; project administration, X.R.; funding acquisition, M.S. All authors have read and agreed to the published version of the manuscript.

**Funding:** This research received no external funding.

**Data Availability Statement:** Data is contained within the article.

**Acknowledgments:** First of all, I would like to show my deepest gratitude to Baotou Research Institute of Rare Earths for providing the necessary laboratory, experimental materials, funds and other support during my doctoral period. Secondly, I shall extend my thanks to Zhao (Dongyue Zhao) for his professional guidance on this article. Finally, I would also like to my family for their encouragement and support.

**Conflicts of Interest:** The authors declare no conflict of interest.

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
