# Peer review of "Insights into Synergy of Copper and Acid Sites for Selective Catalytic Reduction of NO with Ammonia over Zeolite Catalysts"

_catalysts, doi:10.3390/catal13020301_

Round 1
Reviewer 1 Report
This manuscript that a comprehensive interpretation of acid content and metal species in NH3-SCR is provided by considering similar Cu and Al content with respect to the active metal species and acid point of zeolite in NH3-SCR. On the other hand, the discussion from the experimental results was insufficient for the conclusions. This report could be considered fully worthy of submission by modifying the points shown how.
1. There is no quantitativity among Cu-SSZ-13, CuZSM-5 and Cu-BEA in the amount of Cu in the XPS. (Because the sample amount in the measurement area is not constant where the electron beam is applied.) The authors need to change Table 3.
2. When discussing deactivation in the high-temperature SCR performance, there is a need to measure the stability of the zeolite framework after the NH3-SCR reaction. The author should perform XRD measurements on the samples after the reaction.
3. The authors attributed the weak Lewis acid to a surface hydroxyl group, although that hydroxyl group has not been fully observed by IR and 29Si MAS NMR. However, the weak Lewis acid could be attributed to NH3 species adsorbed on NH3. (M. Niwa, M. Iwamoto and K. Segawa: Bull. Chem. Soc.Jpn. 59, 3735 (1986))The authors should reconsider in light of this paper.
4. In NH3 adsorption, the adsorption of the Brønsted acid amount of CuZSM-5 is significantly lower than the Al content (Si/Al=14, around 1.1 mmol g-1). The correlation between kind of acid and NH3-SCR activity is an important claim of the authors, so it is necessary to discuss for this reason.
Author Response
Dear Editors and Reviewers:
Thanks you for your letter and for the reviewers’comments concerning our manuscript entitled “Insights into synergy of copper and acid sites for selective catalytic reduction of NO with ammonia over zeolite catalysts”(Manuscript ID: catalysts-2173267).Those comments are all valuable and very helpful for revising and improving our paper, as well as the important guiding significance to our researches. We have studied comments carefully and have made correction which we hope meet with approval. Revised portion are marked in red in the paper. The main corrections in the paper and the responds to the reviewers’comments are as follows:
Response to Reviewer 1 Comments
Point 1: There is no quantitativity among Cu-SSZ-13, Cu-ZSM-5 and Cu-BEA in the amount of Cu in the XPS. (Because the sample amount in the measurement area is not constant where the electron beam is applied.) The authors need to change Table 3.
Response 1: We have made correction according to the Reviewer’s comments (See the paper, the modification has been marked in red). Since the content of Table 3 is the same as that of Figure 6b, Figure 6b is deleted.
Point 2: When discussing deactivation in the high-temperature SCR performance, there is a need to measure the stability of the zeolite framework after the NH3-SCR reaction. The author should perform XRD measurements on the samples after the reaction.
Response 2: According to the comments of the reviewer, we have added Figure 2c and relevant contents in XRD (See the paper, the modification has been marked in red).
Point 3: The authors attributed the weak Lewis acid to a surface hydroxyl group, although that hydroxyl group has not been fully observed by IR and 29Si MAS NMR. However, the weak Lewis acid could be attributed to NH3 species adsorbed on NH3. (M. Niwa, M. Iwamoto and K. Segawa: Bull. Chem. Soc.Jpn. 59, 3735 (1986))The authors should reconsider in light of this paper.
Response 3: According to the relevant literatures recommended by the reviewer, We have reconsidered the types and distribution of acid sites. First, we have adjusted the sequence of DRIFTS-NH3 and NH3-TPD, and the NH3-DRIFT spectra was re-analyzed and the contents of this section were completely revised. We have found all the characteristic peaks representing Lewis and Bronsted acid sites in the DRIFTS-NH3 spectra, and redefined these two acid sites. The specific content is Four Bønsted acid sites exist on the zeolite catalysts surface , namely Si-OH, Cu-OH, Al-OH and Al-OH-Si, and the Lewis acid sites are mainly two types of Cu exchange sites, which are isolated copper ions in CHA cage and 6-MR. Secondly, according to the DRIFTS-NH3 analysis results, we redistributed the acid sites corresponding to different NH3 desorption temperatures in the NH3-TPD analysis and revised Figure 10b and related contents (See the paper, the modification has been marked in red).
Point 4: In NH3 adsorption, the adsorption of the Brønsted acid amount of CuZSM-5 is significantly lower than the Al content (Si/Al=14, around 1.1 mmol g-1). The correlation between kind of acid and NH3-SCR activity is an important claim of the authors, so it is necessary to discuss for this reason.
Response 4: We have added Figure 10c in NH3-TPD analysis to show relationship between acid sites and catalytic activity (See the paper, the modification has been marked in red).
We tried our best to improve the manuscript and made some changes in the manuscript. These changes will not influence the content and framework of the paper. And here we did not list the changes but marked in red in revised paper. We appreciate for Editors/Reviewers’ warm work earnestly, and hope that the correction will meet with approval.
Once again, thank you very much for your comments and suggestions.
Yours sincerely,
Wenyi Zhao
Reviewer 2 Report
The manuscript reports the function of copper sites in Cu-SSZ-13, Cu-ZSM-5 and Cu-Beta catalysts on the NH3-SCR reaction. These catalysts were characterized using proper techniques, and the relationship between the active sites and catalytic activity was established. The results are comprehensive and interesting in the NH3-SCR research field. However, some minor issues must be taken into account before the manuscript is accepted.
1. NOx conversion in the figures and formula should be changed to NO conversion. 2. In this paper, the deterioration rate is used to express the durability of catalyst. Please explain the basis for use. 3. In DRIFTS-NH3 adsorption experiments (Figure 10), the NH3 bands in the 1400-1650 cm-1 region is not all Bronsted acid sites in the catalysts, and the band at 3540cm-1 is also a feature of the Brønsted acid sites. Therefore, the description of "The intensity sequence of the absorption peaks at 1400-1650cm-1 is Cu-SSZ-13 > Cu-Beta > Cu-ZSM, which shows that the amount of the strong acid sites (Brønsted acid sites) of the Cu-SSZ-13 is obviously superior to that of Cu-Beta and Cu-ZSM, which is consistent with the results of NH3-TPD." in the text is not accurate, so the analysis here should be removed or compared again. 4. The usage of English should be polished throughout the manuscript.
Author Response
Dear Editors and Reviewers:
Thanks you for your letter and for the reviewers’comments concerning our manuscript entitled “Insights into synergy of copper and acid sites for selective catalytic reduction of NO with ammonia over zeolite catalysts”(Manuscript ID: catalysts-2173267).Those comments are all valuable and very helpful for revising and improving our paper, as well as the important guiding significance to our researches. We have studied comments carefully and have made correction which we hope meet with approval. Revised portion are marked in red in the paper. The main corrections in the paper and the responds to the reviewers’comments are as follows:
Response to Reviewer 2 Comments
Point 1: NOx conversion in the figures and formula should be changed to NO conversion.
Response 1: According to the reviewer's suggestion, We have changed the NOx conversion in the text, figure and formula into NO conversion.
Point 2: In this paper, the deterioration rate is used to express the durability of catalyst. Please explain the basis for use.
Response 2: the durability and stability of the catalysts are expressed by deterioration rate, which was the average value of NO conversion decrease of aging samples at four temperature test points (200 ℃, 250 ℃, 450 ℃ and 500 ℃). The test method of deterioration rate is mainly used for durability test of bench test (see Standards of China Environmental Protection Industry Association T/CAEPI 12.2-2017 for test method).
Point 3: In DRIFTS-NH3 adsorption experiments (Figure 10), the NH3 bands in the 1400-1650 cm-1 region is not all Bronsted acid sites in the catalysts, and the band at 3540cm-1 is also a feature of the Brønsted acid sites. Therefore, the description of "The intensity sequence of the absorption peaks at 1400-1650cm-1 is Cu-SSZ-13 > Cu-Beta > Cu-ZSM, which shows that the amount of the strong acid sites (Brønsted acid sites) of the Cu-SSZ-13 is obviously superior to that of Cu-Beta and Cu-ZSM, which is consistent with the results of NH3-TPD." in the text is not accurate, so the analysis here should be removed or compared again.
Response 3: First, we have adjusted the sequence of DRIFTS-NH3 and NH3-TPD. Secondly,the DRIFTS-NH3 spectra was re-analyzed and the contents of this section were revised. We have found all the characteristic peaks representing Lewis and Bronsted acid sites, and redefined these two acid sites. The specific content is Four Bønsted acid sites exist on the zeolite catalysts surface , namely Si-OH, Cu-OH, Al-OH and Al-OH-Si, and the Lewis acid sites are mainly two types of Cu exchange sites, which are isolated copper ions in CHA cage and 6-MR. Finally, Because the DRIFTS-NH3 can only be used for semi-quantitative analysis of acid sites, the NH3-TPD can accurately calculate the acid amount. Therefore, in DRIFTS-NH3 analysis, the comparison of the amount of acid sites of these three catalysts is no longer carried out, but the quantitative analysis is mainly carried out in NH3-TPD.
Point 4: The usage of English should be polished throughout the manuscript.
Response 4: According to the reviewer's suggestion, our manuscript has been revised and adjusted in language.
We tried our best to improve the manuscript and made some changes in the manuscript. These changes will not influence the content and framework of the paper. And here we did not list the changes but marked in red in revised paper. We appreciate for Editors/Reviewers’ warm work earnestly, and hope that the correction will meet with approval.
Once again, thank you very much for your comments and suggestions.
Yours sincerely,
Wenyi Zhao
Reviewer 3 Report
In this manuscript, three Cu catalysts loaded on SSZ-13, ZSM-, and Beta were compared and analyzed in related to porosity, morphology, Cu species, redox properties, acidity, NO adsorption, NO+O2 adsorption, and so on. The activity and hydrothermal aging durability were measured. The relationship between Cu species, acidity, and activity was established.
Overall, the manuscript was well drafted and analyzed. Thus, the reviewer suggests to accept it after a very minor revision. Here are a few comments.
1. China should be capitalized.
2. The reviewer wonder, why did the authors choose a Beta zeolite with so low crystallinity and so high mesoporous volume. As the crystallinity highly affect the total pore volume, particle size, stability, and acidity, which will further affect the Cu species and catalytic activity.
2. For H2-tpr calculation and fraction of various Cu species, the reduction of Cu2+ to Cu+ and further to Cu0 is a cascade process, rather than parallel process. So the calculation method should be clarified.
3. For DRIFT spectra of NO adsorption, blank samples without copper should be provided.
4. For Brønsted and Lewis acid sites, the pyridine adsorption test is more common. So the reviewer suggest to provide the pyridine IR spectra at least for the best catalyst: Cu-SSZ-13. Meanwhile, the recent paper about complete acidity analysis of zeolites should be cited (doi.org/10.1016/j.jcat.2019.05.020).
Author Response
Dear Editors and Reviewers:
Thanks you for your letter and for the reviewers’comments concerning our manuscript entitled “Insights into synergy of copper and acid sites for selective catalytic reduction of NO with ammonia over zeolite catalysts”(Manuscript ID: catalysts-2173267).Those comments are all valuable and very helpful for revising and improving our paper, as well as the important guiding significance to our researches. We have studied comments carefully and have made correction which we hope meet with approval. Revised portion are marked in red in the paper. The main corrections in the paper and the responds to the reviewers’comments are as follows:
Response to Reviewer 3 Comments
Point 1: China should be capitalized.
Response 1: We have changed "china" to "China” in the introduction and corrected other similar errors in the paper.
Point 2: The reviewer wonder, why did the authors choose a Beta zeolite with so low crystallinity and so high mesoporous volume. As the crystallinity highly affect the total pore volume, particle size, stability, and acidity, which will further affect the Cu species and catalytic activity.
Response 2: First of all, our laboratory mainly develops H-SSZ-13, while H-Beta and H-ZSM-5 catalysts are purchased from outside. Secondly, due to lack of experience and comparison of multiple products, the latter two catalysts were directly analyzed. Finally, we will choose catalysts more rigorously and carefully in the future to make the experimental data more comparable.
Point 3: For H2-TPR calculation and fraction of various Cu species, the reduction of Cu2+ to Cu+ and further to Cu0 is a cascade process, rather than parallel process. So the calculation method should be clarified.
Response 3: According to the reviewer's suggestion, first of all, we added the reduction relationship between Cu2+and Cu+ ions in the zeolite catalysts in the H2-TPR analysis. The added content is as follows: the reduction of isolated Cu2+ ions located at the ion exchange sites often need to go through two steps. First, the isolated Cu2+ ions are reduced to Cu+ ions at a lower temperature, and then the Cu+ ions are completely reduced to Cu0 at a higher temperature. These Cu+ ions include the intermediate products of the two-step reduction of isolated Cu2+ ions and the Cu+ ions existing in the catalyst. Secondly, the ratio of various Cu ions to total Cu was calculated according to the H2 reduction peak areas and used to measure the concentration of Cu ions in the catalysts.
Point 4: For DRIFT spectra of NO adsorption, blank samples without copper should be provided.
Response 4: According to the suggestion of the reviewer, we carried out the blank experiment of H-type zeolite, but no characteristic peak of NO was found in the 1650-2150 cm-1 region. In addition, we have also consulted the relevant literatures of DRIFTS-NO, and there is almost no introduction of NO characteristic peaks outside the 1650-2150 cm-1 region. Therefore, we believe that there is no addition if it is not comparable with copper based zeolites.
Point 5: For Brønsted and Lewis acid sites, the pyridine adsorption test is more common. So the reviewer suggest to provide the pyridine IR spectra at least for the best catalyst: Cu-SSZ-13. Meanwhile, the recent paper about complete acidity analysis of zeolites should be cited.
Response 5: First of all, I'm very sorry that because of the Chinese New Year holiday and winter holiday and our laboratory does not have the test conditions for pyridine IR , the test school we contacted needs to wait until the middle and late February to resume the test, so this experiment cannot be completed in a short time. Secondly, in order to better identify the Bronsted and Lewis acid sites in the zeolites, and we have consulted a large number of literatures on the characterization of acid sites (including pyridine IR), so that we have a deeper understanding of the types and distribution of acid sites. Although pyridine IR is more common, DRIFTS-NH3 can also well reflect the presence of Bronsted and Lewis acid sites. Therefore, we have adjusted the sequence of DRIFTS-NH3 and NH3-TPD, and the DRIFTS-NH3 spectra was re-analyzed and the contents of this section were completely revised. We have found all the characteristic peaks representing Lewis and Bronsted acid sites in the DRIFTS-NH3 spectra, and redefined these two acid sites. The specific content is Four Bønsted acid sites exist on the zeolite catalysts surface, namely Si-OH, Cu-OH, Al-OH and Al-OH-Si, and the Lewis acid sites are mainly two types of Cu exchange sites, which are isolated copper ions in CHA cage and 6-MR (See the paper, the modified content has been marked in red). Thirdly, we have added three new papers on the acidity analysis of zeolite. Finally, I will add pyridine IR characterization in my doctoral thesis to enrich the experimental content and better and more scientific reaction experimental conclusions.
We tried our best to improve the manuscript and made some changes in the manuscript. These changes will not influence the content and framework of the paper. And here we did not list the changes but marked in red in revised paper. We appreciate for Editors/Reviewers’ warm work earnestly, and hope that the correction will meet with approval.
Once again, thank you very much for your comments and suggestions.
Yours sincerely,
Wenyi Zhao
Round 2
Reviewer 1 Report
This manuscript provides a comprehensive interpretation of acid content and metal species in NH3-SCR, considering similar Cu and Al contents with respect to active metal species and acid points in zeolites in NH3-SCR. The relationship between acid quality and catalytic activity is also analyzed in detail based on NH3-TPD and IR. The manuscript can be accepted without additional revisions.